# Influence of the Loading Speed on the Ductility Properties of Corroded Reinforcing Bars in Concrete

**DOI:** 10.3390/ma12060965

**Published:** 2019-03-22

**Authors:** Angela Moreno Bazán, María de las Nieves González, Marcos G. Alberti, Jaime C. Gálvez

**Affiliations:** 1Departamento de Ingeniería Civil: Construcción, E.T.S de Ingenieros de Caminos, Canales y Puertos, Universidad Politécnica de Madrid, C/Profesor Aranguren, s/n, 28040 Madrid, Spain; angela.moreno@upm.es (A.M.B.); marcos.garcia@upm.es (M.G.A.); 2Departamento de Construcciones Arquitectónicas y su Control, E.T.S de Edificación, Universidad Politécnica de Madrid, Avda/Juan de Herrera, 6, 28040 Madrid, Spain; mariadelasnieves.gonzalez@upm.es

**Keywords:** corrosion, ductility, mechanical properties, reinforced concrete, tensile strength, equivalent steel

## Abstract

In this work 144 reinforcing bars of high-ductility steel named B500SD were subjected to an accelerated corrosion treatment and then tested under tension at different loading speeds in order to assess the effect of corrosion on the ductility properties of the rebars. Results showed that the bars with a corrosion level as low as the one reducing the steel mass by 1% gave rise to a significant degradation on the ductility properties when a high loading speed was applied in tensile tests. In that case, the equivalent steel concept is useful to reduce the destabilising effect. Thus, the research significance lies in the assessment of the influence of the loading speed at which the tensile test is performed for the reinforcement bars that largely depends of the ductility criteria used.

## 1. Introduction

Corrosion of steel rebars is one of the most common deterioration mechanisms identified in reinforced concrete structures. Such corrosion, whether induced by carbonation, chlorides or another attack, affects the overall serviceability and durability of the structure with consequences such as a reduction of the effective cross-section of the steel rebars, cracking and spalling of the concrete cover and the degradation of bond strength [1,2,3].

In the case of chloride corrosion, the attack is mostly local and commonly known as pit corrosion. These chlorides can be found in sea water, industrial wastewater, and, among others, deicing salts [4]. Corrosion occurs after the depassivation of the alkaline barrier when sufficient oxygen and moisture are available. Then the passive film is locally destroyed and a process of local corrosion is initiated. The crucial amount of chlorides needed to interrupt the passive cover is 0.4%–1% by mass of cement as an appropriate chloride threshold [5].

Over the last years more and more qualitative and quantitative research has been carried out about the reduction of the steel bars effective cross-section areas through tensile tests run by chloride corrosion. The Spanish Structural Concrete Code EHE-08 and the Eurocode EC-2 [6,7] require limited values in the mechanical properties of high-ductility steel in terms of both strength and strain. The rebar strength has a significant influence on the structural strength of concrete reinforced members and the codes require minimum values for the steel yield strength and maximum tensile strength. Additionally, due to the consideration of dynamic and seismic actions, consideration of properties in relation to the steel ductility is also required. It is essential that an effective method to reflect the relationship between the mechanical and ductility property of steel bars and the corrosion be found [8,9,10,11].

One way in which ductility can be considered is in relationship with the fracture energy that is the area covered by the strain-stress curve. This energy depends on the plastic deformation capacity of steel up to breaking point. The higher the area, the higher is the capacity of steel to dissipate energy under dynamic loads. In addition, for dynamic and impact loads, the speed at which the load is applied is important.

The Codes EHE-08 and EC-2 require that the steel bars meet ductility properties based on total elongation at maximum force (*A*_gt_) and the ratio between tensile strength and yield strength (*R*_m_/*R*_e_). Table 1 shows the limits of those parameters as required by the EHE-08 code in order to qualify the steel as of high ductility according to standard UNE 36065:2011 [12]. In addition, although the percentage elongation after fracture (*A*_u,5_) is included in this code, it is not considered in other codes. 

The conventional approach to steel rebar corrosion considers a reduction in the area of the bar section proportional to the degree of corrosion. Most of the published works [13,14,15,16,17] report systematic reduction of strength and strain at maximum load when the degree of corrosion increases. Recent studies [18] have shown that corrosion takes place in local spots of the bar surface (pitting), a weakening of strength occurs at these spots (a notch effect), and the bar strength falls under the minimum values required by the codes, even with very small degrees of corrosion. Nevertheless, the reduction of strain is greater than the loss of strength in the bar [19].

With low levels of corrosion the loss of strength is also low: this means that the structural elements can still meet their resistance function, though the reduction of strain may not meet the minimum values required in Table 1 to ensure enough ductility. Previous studies [20,21] have shown that the ratio *R*_m_*/R*_e_ remains constant with the increase of the corrosion level. This means that the steel may amply meet the *R*_m_*/R*_e_ requirement but not the requirement of *A*_gt_. 

In these cases, the use of the equivalent steel concept as a ductility criterion based on both *R*_m_*/R*_e_ and *A*_gt_ may be highly useful [22,23]. Table 2 shows the minimum values obtained with the EHE-08 requirements in application of the equivalent steel formulas [24,25,26,27,28] as proposed by Cosenza (*p*), Creazza (*A**) and Ortega (*Id*).

Moreover, although the standard test procedure of tensile tests of reinforcing bars ISO 15630-1:2010 [29] is remarkably complete, no recommendation regarding the loading speed for the tensile test is set. This might have a significant impact on the test results. For this reason, in this study the variations of the mechanical properties of steel rebars as a function of the degree of corrosion and the loading speed applied in the tensile tests are reported. For such a purpose, 144 12-mm diameter high-ductility steel rebars, named B500SD, were tested in tension after an accelerated corrosion treatment, embedded in NaCl contaminated concrete. The results showed that the influence of high-speed loading is significant if only the Codes EHE-08 and EC-2 are used. In order to establish a better relationship between the tensile test and the minimum effective cross-sectional area, the use of the equivalent concept is required.

## 2. Materials and Methods 

### 2.1. Materials

Twelve concrete slabs of 30 × 40 × 10 cm^3^ were fabricated, with each one having 12 reinforcing bars partially embedded, making a total of 144 bars to be corroded and tested in tension (Figure 1). Table 3 shows the concrete composition. The mean compression strength of the concrete was 26 MPa. The steel type was B500SD, used normally in structures in seismic areas due to the higher properties in terms of ductility.

River silica, sand, round gravel, and cement CEM II/A-L 32.5, according to standard RC-16 [30] were the mix basic materials. The water/binder ratio was 0.6. CaCl_2_ with a concentration of 3% relative to the weight of cement was diluted in the mixing tap water in order to destroy the passive state of the reinforcing bars. After casting and demoulding, the slabs were cured for 28 days in chamber under ambient conditions at 25 °C and 99% of relative humidity. In order to avoid corrosion initiation and propagation at the point where the bars protrude the concrete slab, each bar was wrapped with insulating tape in these points for lengths of 3 cm both outside and inside the concrete.

### 2.2. Accelerated Corrosion

Because natural corrosion of the steel bars is a time-consuming process, an artificial accelerated test by an electrochemical method was utilised to simulate the corrosion process [31]. Thus, corrosion was accelerated by imposing an electrical current between the reinforcement, the working-electrode and a counter-electrode, in a way that oxidation is enforced. The flow of the current was established by an external constant intensity. Nevertheless, a corrosion pit was found on the steel bars by electrochemical method that showed that non-uniform corrosion occurred too [32,33]. Therefore, the uniformity of corrosion was unconnected with the corroded method, but rather with the degree of corrosion. Many researchers have already used this technique to gather information about corrosion processes in reinforced concrete [34,35]. In this test, the accelerated corrosion process was activated by an electrical current by the application of a constant anodic current between the bars and a lead plate placed on top of the slabs, acting as the cathode. A soaked textile pad placed between the concrete slab and the lead plate ensured an even distribution of the electric current.

The value of the current in each bar was controlled by means of a digital multimeter, involving periodical recording of the voltage and adjustment of the electrical potential at the power source to ensure a constant current value of approximately 10 μA/cm^2^ in each bar. This accelerated corrosion test is only valid if the intensity remains rather low (<200 μA/cm^2^) with respect to Faraday’s law. Otherwise a significant increase of strain response, and consequently the crack width, will occur. Densities can reach 200 μA/cm^2^, with increasingly more appearing that will undermine accuracy of the experiment. These current values are broadly explained by Andrade [36], Maaddawy [37] or Suvash [38]. In order to obtain distinct corrosion levels, the current was disconnected at different ages after cracks appeared in the concrete slabs (Figure 2).

Once the corrosion process was over, the slabs were demolished, the oxide and cement of each bar surface were mechanically cleaned by a brush according to standard ASTM G1-03 (2017) [39]. Then, if such a treatment could not eliminate all the corrosion products, the oxide was cleaned by a chemical process by immersing the bars in a bath for 10 min. It was then rinsed with ethanol and water and dried according to standard ISO 8407 [40]. As it can be observed in Figure 3, the corrosion was extended throughout the whole surface of embedded bars.

The degree of corrosion or corrosion level (*Q*_corr_) was measured by the gravimetric procedure of weighing the bars after the full cleaning of all corrosion products. The gravimetric cross-sectional loss of the corroded steel reinforcement was deduced by Equation (1) [41].

(1)$$Q_{\mathrm{corr}}\;=\;\frac{m_{0}-m_{\mathrm{res}}}{m_{0}}\times 100$$ where *Q*_corr_ is the corrosion degree of the steel reinforcement in percentage and $\;m_{0}$ and $m_{\mathrm{res}}$ are the mass of initial reinforcement corresponding to the portion of rebar that participates, respectively, in corrosion and the residual mass of the corroded steel reinforcement portion.

In order to calculate the residual diameter value of the corroded bar cross-section, the residual mass of the corroded steel reinforcement is used and determined following the equivalent section definition [14,42].

Thus, the residual diameter of the corroded bars was computed by Equation (2) as follows:(2)$$\emptyset_{\mathrm{res}}\;=\;\sqrt{\frac{4m_{\mathrm{res}}\;}{\pi \;7.85\;L_{c}}}$$ where *L_c_* is the corroded length of the bar (cm) and 7.85 is the specific weight of steel (g/cm^3^).

### 2.3. Strength Tests

Once the corrosion process was finished and the bars free of all corrosion products, the bars were tested under tension according to standard EN ISO 7500-1:2018 [43] by a multitest IBERTEST press with a loading capacity of 100 kN, controlled by the WINTest 32 software program. The strain measurements were performed with an extensometer 2-IBER-25 with a 50 mm base.

Before the test, for the manual determination of the elongation after fracture (Equation (3), the entire bar received fine marks with multiples of 5mm following the UNE EN ISO 6892-1:2017 standard [44]. In addition, for the total elongation at maximum force a class two strain gauge, according to the ISO 9513:2012 standard, was used [45].

(3)$$A_{u,5}\;=\;\frac{L_{u}-L_{0}}{L_{0}}\times 100$$ where *A*_u,5_ is the permanent elongation of the gauge length expressed as a percentage of the original gauge length, *L*_u_ is the final gauge length after fracture and *L*_0_ is the original gauge length. 

The tests were controlled in terms of load when the bar behaved in the elastic range and deformation when the test was running in the plastic zone at three different speeds. The standard speed *V*_m_ (medium loading speed) was that recommended by standard UNE-EN ISO 15630-1:2011 [29], that is to say, 3.7 kNs and 20.1 mm/min. A speed three times faster *V*_h_ (high loading speed) with 11.1 kN/s and 60.3 mm/m, and a speed *V*_l_ (low loading speed) three times slower with 1.23 kN/s and 6.7 mm/min were also used.

A designating code B-*XXX*-*Y* was used to identify each bar sample, with B meaning bar, *XX* the bar number (1 to 144) and *Y* the loading speed (low, medium or high). The (*) in the nomenclature shows that the rebar did not meet certain EHE-08 requirements for B500SD steel.

## 3. Results and Discussion

Based on the test results, the values of the equivalent steel ductility parameters as per Ortega (*Id*), Cosenza (*p*) and Creazza (*A**) were calculated for each bar sample (see Equations (4)–(6)). Table A1, Table A2 and Table A3 in Appendix A show the results and the ductility parameters for, respectively, low, medium and high loading speed. The level of corrosion was quantified by *Q*_corr_ as per Equation (1) and used to order the table lists. 

The results include the yield strength (*R*_e_), tensile strength (*R*_m_), the total elongation at maximum force (*A*_gt_) and the permanent elongation of the gauge length *A*_u,5_. All mechanical properties were calculated with respect to the residual diameter of the corroded bars ($\emptyset_{\mathrm{res}}$ in Equation (2)). In addition, the equivalent steel ductility parameters were calculated with respect to Equations (4)–(6).

(4)$$Id\;=\;1+\;\left(1+\frac{R_{m}}{R_{e}}\right)\left(\frac{A_{\mathrm{gt}}}{\varepsilon_{y}}-1\right)$$

(5)$$A^{*}\;=\;\frac{\;2}{3}\left(R_{m}-R_{y}\right)\left(A_{\mathrm{gt}}-\varepsilon_{\mathrm{sh}}\right)$$

(6)$$p\approx \;A_{\mathrm{gt}}{}^{0.75}{\left(\frac{R_{m}}{R_{e}}-1\right)}^{0.9}$$

Those bars with lower values of mechanical parameters than those required for steel B500SD in EHE-08 have been highlighted with an asterisk in Table A1, Table A2 and Table A3 (see Appendix A). The maximum and minimum values for the parameters can be seen in Table 4.

A general overview of the values in Table 4 reveals that the high loading speeds (*V*_h_) cause a distorting of the results, with yield strength and tensile strength values being significantly lower than it expected. Thus, the equivalent steel ductility parameters at this loading speed would be highly recommended. As the values of total elongation at maximum force declined substantially, in the majority of the cases three times less than the total elongation recorded for the control, corrosion is more sensitive to strain than to stress. 

Table 5 shows the yield strength and tensile strength for the three loading speeds of steel bars at four levels of corrosion and the percentage of strength loss of yield and tensile strength with 1%, 2%, 3% and 4% of corrosion degree and yield and tensile strength without corrosion, following Equation (7).
(7)$$\mathrm{Strength}\;\mathrm{reduction}\;\mathrm{rates}\;=\left(\frac{\left(\frac{R_{\mathrm{ei}}}{R_{e0}}\right)+\left(\frac{R_{\mathrm{mi}}}{R_{m0}}\right)}{2}\right)\times 100$$

With a corrosion degree of 1%, reduction rates of yield strength and tensile strength are around 5% and 6% for all the loading speeds. Similar strength reduction rates of approximately 8% and 9% are found in bars when the corrosion level increases to 2%. However, a corrosion level of more than 3% will induce a greater tensile-strength reduction of approximately 16% and 19% for medium and high levels of loading speeds and 13% for a low loading speed. This confirms that the loading speed has a greater influence when the rebars have higher degrees of corrosion. 

Figure 4 shows how these deviations in the results can be studied based on the equivalent steel criteria and how they can be analysed in the results. The figure shows the comparison of the percentage of specimens that meet the various equivalent criteria, at the different loading speeds *V*_l_, *V*_m_ and *V*_h_. 

As expected, when using EHE-08 criteria, the loading speed is important with high levels of corrosion. Even with corrosion rates of up to 1% there is a significant difference for high loading speed (*V*_h_) as compared with low and medium speeds (*V*_l_, *V*_m_). In addition, for corrosion rates of up to 1% all equivalent steel criteria are met for low and medium loading speeds. For high-speed loading the criteria offered by Creazza are scarcely met, although fulfilment is frequent for Cosenza and Ortega criterion. With corrosion rates of higher than 1%, fulfilment of EHE-08 ductility criteria was low, less so for Cosenza criterion and high for the Creazza and Ortega criterion. Thus, in general, the three equivalent steel concepts offered by Cosenza, Creazza and Ortega serve as useful criteria for high loading speeds and corrosion rates under 1%. Given that more that 90% of the bar specimens meet the ductility criteria, the concept is quite advantageous in assessing structural ductility with corroded reinforcement. Regardless of the loading speed considered, the EHE-08 ductility requirements are met by more than 90% of the bar specimens for corrosion rates of up to 1%, though only 20% of the specimens meet such requirements when the corrosion rate is higher than 1%. The main reason is the systematic reduction of the total elongation at maximum load (*A*_gt_) when it increases the corrosion rates [30] up to values that fail the minimum ones required by EHE-08.

Summaries of representative strain-stress curves are plotted in Figure 5, Figure 6 and Figure 7 for, respectively, each low, medium and high loading speed. The numbers in % indicate the corrosion level of the rebar.

It can be seen that with the development of corrosion, the yield strength, tensile strength and total elongation at maximum load decreased under different strain rates. In addition, the yield plateau shortened or even disappeared. In contrast, given that with high-corrosion levels an anomalous elongation of the yield plateau is produced, the loading speed has a significant influence on this area of the curve. Compared with the uncorroded rebars, the decreased yield and tensile strength of the corroded rebars were mainly caused by the reduction of fracture cross-sectional areas. The decreased total elongation and the shortened yield plateau were due to intensified stress concentrations at the corrosion pits [46].

Figure 8 shows the range of deformations (mean value) in the yield zone for corrosion rates of lower than 1% (a) and higher than 1% (b), for the three loading speeds used. It can be seen that deformation is similar for low and medium speeds regardless of the corrosion rate. At high corrosion rates, deformation is much greater for the high loading speed (as discussed above). 

Regardless of the corrosion level, in Figure 9 it is possible to see that the deformation of bar specimens in the yield zone showed less dispersion for low and medium speeds in comparison with high speed.

The evolution of the mechanical properties obtained in the tensile tests as a function of the corrosion rate can be observed in Figure 10. It shows the average cross-section diameter of the specimen $\emptyset_{\mathrm{res}}$ after the corrosion process. Each colour denotes the loading speed. The figure also shows that the yield strength, tensile strength and total elongation at maximum force, regardless of the loading speed, decrease with the increase in corrosion level. The dispersion of the results occurs only with high corrosion levels. Therefore, the loading speed in this test program seemed to have no significant effect on either strength at a low level of corrosion.

If the corrosion had been uniform along the bar, the adjusting trend lines would have been horizontal. However, the lines decrease for all loading speeds [21]. This is because corrosion is not homogeneous in the bar surface, but occurs in a series of pitting spots typical for chloride corrosion of steel [46,47]. In these spots, the cross-section area of the bar is smaller than the average in the test results (see Figure 11). Additionally, corrosion takes place in the outer thickness of the bar surface composed by martensite, a metallographic material produced by the rolling mill when the bar was fabricated. Martensite has higher strength properties (*R*_m_*, R*_e_) than the ferrite composing the internal core of the bar. The destruction of part of this stronger outer layer explains the reduction of the average strength values in the bar cross-section.

The evolution of ratio *R*_m_*/R*_e_ for the three loading speeds is shown in Figure 12. Regardless of the loading speed, the ratio has a small fluctuation near a fixed value while the corrosion ratio changed from zero to more than 4%. It is indicated that the ratio of *R*_m_*/R*_e_ is irrelevant to the average corrosion ratio which shows that the decrease of yield strength is induced by the decrease of effective cross-section area of the steel bars [9,21].

Figure 13 shows the evolution of the three ductility parameters based on the steel equivalent concept as a function of the corrosion rate for the three loading speeds. All parameter values decrease when the corrosion rate increases regardless of the loading speed. Parameters evolve similarly for low and medium loading speeds (parallel lines). However, the degree of scatter with high-speed loading would not provide accurate conclusions.

## 4. Conclusions 

The main conclusions can be summarised as follows:With the exception of the ratio *R*_m_*/R*_e_ there is a systematic reduction of all strength and durability parameters (*R*_m_, *R*_e_, *A*_gt_, *p, A** and *Id*) for increasing corrosion rates.The increasing of the corrosion level leads to modification of the stress-strain diagram, losing the yield plateau and showing cold-drawn behaviour.The loading speed of the tensile test is a variable which, along with the corrosion rate, governs the values and the evolution of all of the studied parameters (except the ratio *R*_m_*/R*_e_).The equivalent steel concept is useful to evaluate the ductility behaviour of corroded reinforcement bars in concrete regardless of the loading speed in the tensile test.The higher the tensile test loading speed, the higher is the yield zone in the strain–stress relationship curve.With corrosion rates as low as 1%, there is a change in the strain-stress curve which means that in some cases the yield plateau disappears and the steel behaves as a cold-formed steel.Finally, the research significance lies in the assessment of the influence of the loading speed at which the tensile test is performed for the reinforcement bars that largely depends of the ductility criteria used.

## Figures and Tables

**Figure 1 materials-12-00965-f001:**
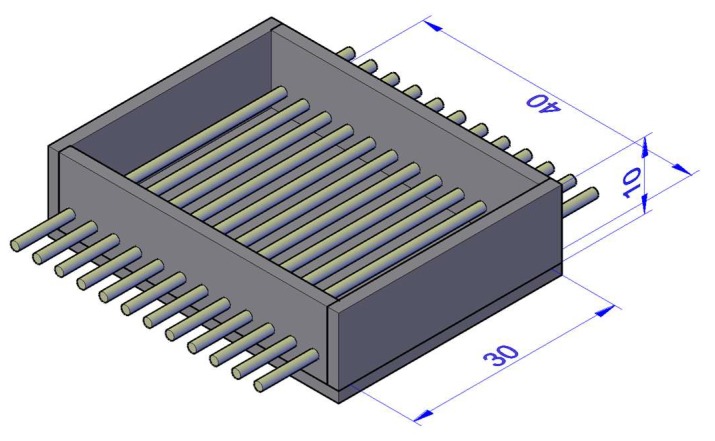
Setup used for the fabrication of slabs (30 × 40 × 10 cm^3^).

**Figure 2 materials-12-00965-f002:**
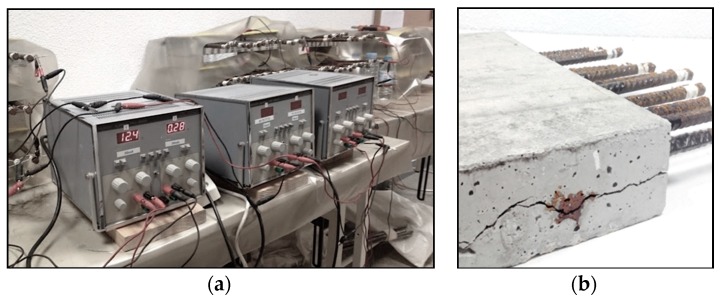
Electrical connection of the bars for the accelerated corrosion process (**a**) and cracks in the concrete slabs after disconnecting and ready for demolition (**b**).

**Figure 3 materials-12-00965-f003:**
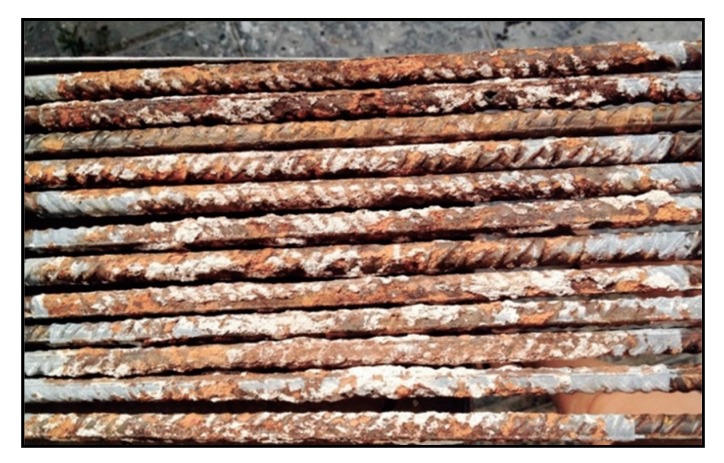
Rebars after the first treatment according to standard ASTM G1-03 (2017) [39].

**Figure 4 materials-12-00965-f004:**
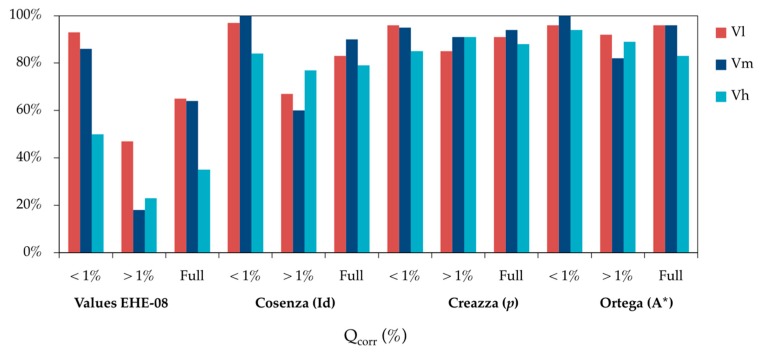
Comparison of the percentage of bar specimens that meet different ductility criteria in tensile strength tests run at different loading speeds: low *V*_l_, medium *V*_m_ and high *V*_h_.

**Figure 5 materials-12-00965-f005:**
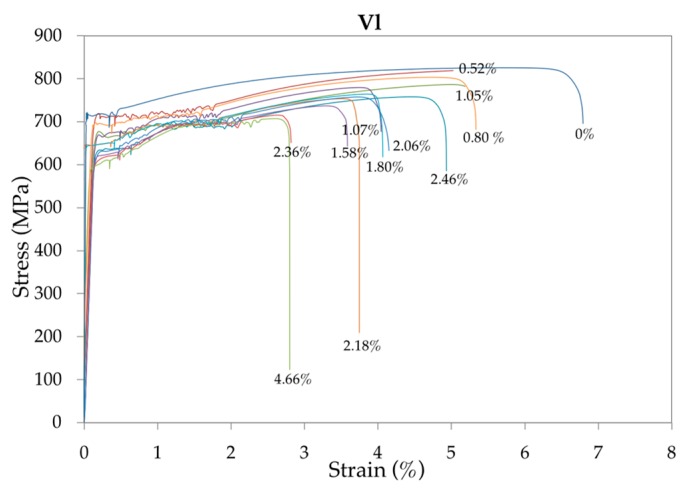
Representative summary of the strain-stress curves of the 48 bar specimens tested at low speed *V*_l_ for increasing corrosion rates.

**Figure 6 materials-12-00965-f006:**
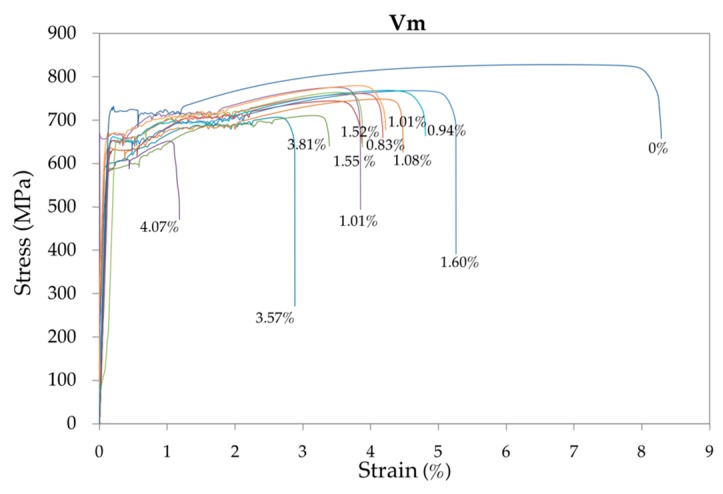
Representative summary of the strain-stress curves of the 48 bar specimens tested at medium (standard) speed *V*_m_ for increasing corrosion rates.

**Figure 7 materials-12-00965-f007:**
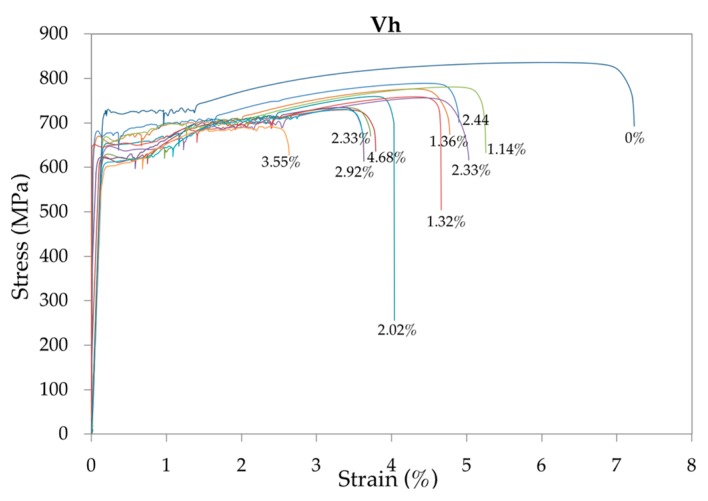
Representative summary of the strain-stress curves of the 48 bar specimens tested at high speed *V*_h_ for increasing corrosion rates.

**Figure 8 materials-12-00965-f008:**
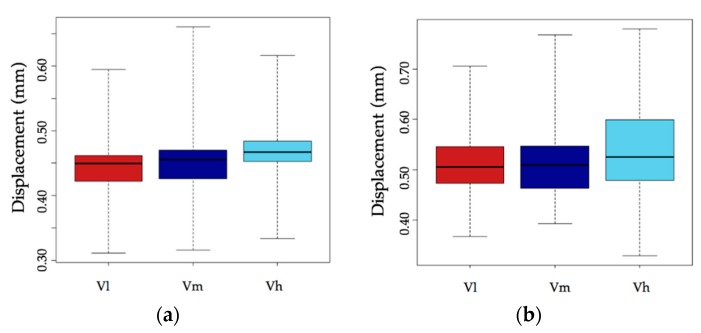
Deformation of bar specimens in the yield zone for corrosion rates under 1% (**a**) and over 1% (**b**) and for the three loading speeds *V*_l_, *V*_m_ and *V*_h_.

**Figure 9 materials-12-00965-f009:**
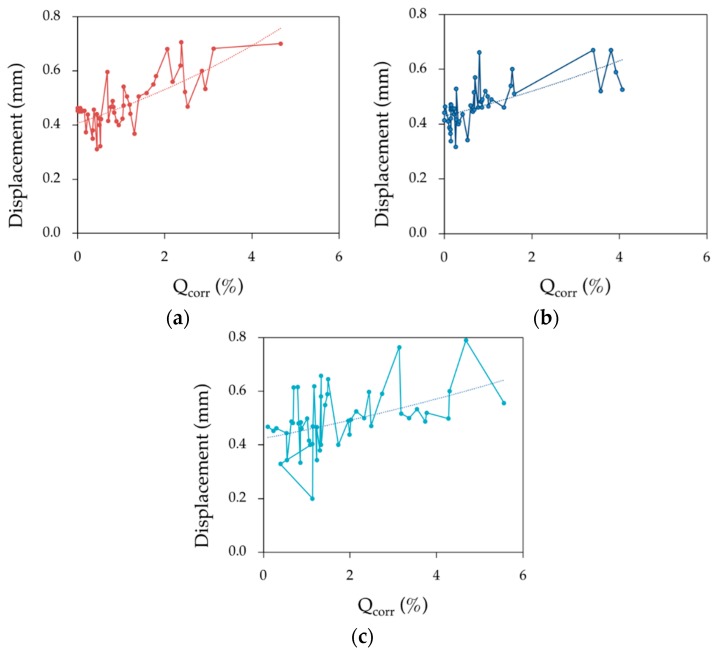
Deformation of bar specimens in the yield zone as a function of the loading speed *V*_l_ (**a**), *V*_m_ (**b**) and *V*_h_ (**c**).

**Figure 10 materials-12-00965-f010:**
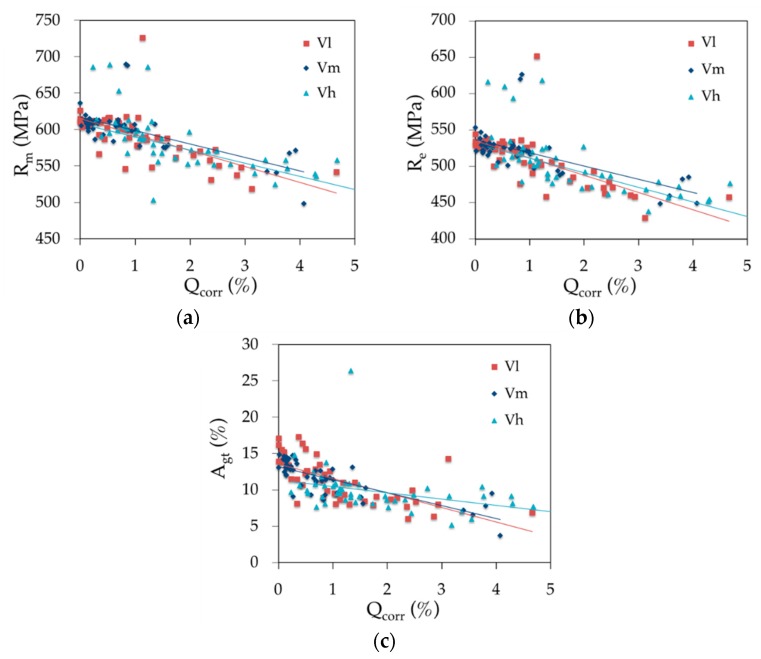
Effects of the corrosion rate and loading speed on the (**a**) tensile strength *R*_m_, (**b**) yield strength *R*_e_ and (**c**) total elongation at maximum force *A*_gt_.

**Figure 11 materials-12-00965-f011:**
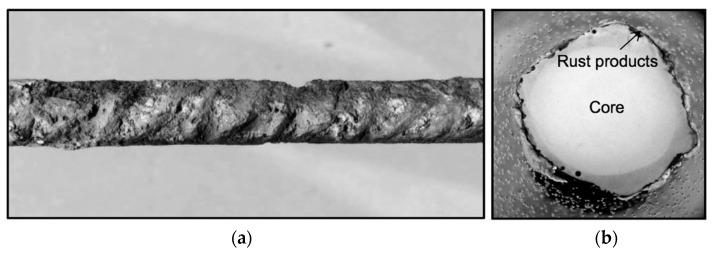
Microscopy image of the corroded surface (**a**) of bar specimen B087M with *Q*_corr_ = 1.07% and the cross-section (**b**).

**Figure 12 materials-12-00965-f012:**
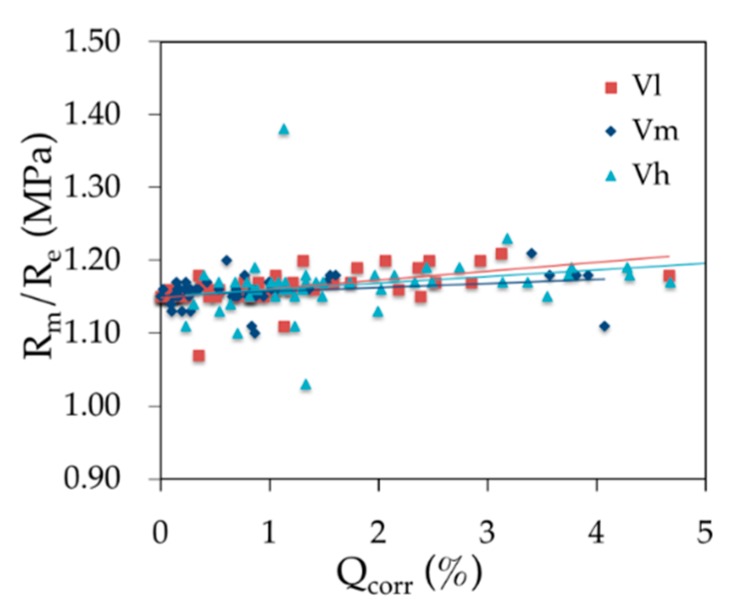
Effects of the corrosion rate and the loading speed on the ratio *R*_m_*/R*_e_.

**Figure 13 materials-12-00965-f013:**
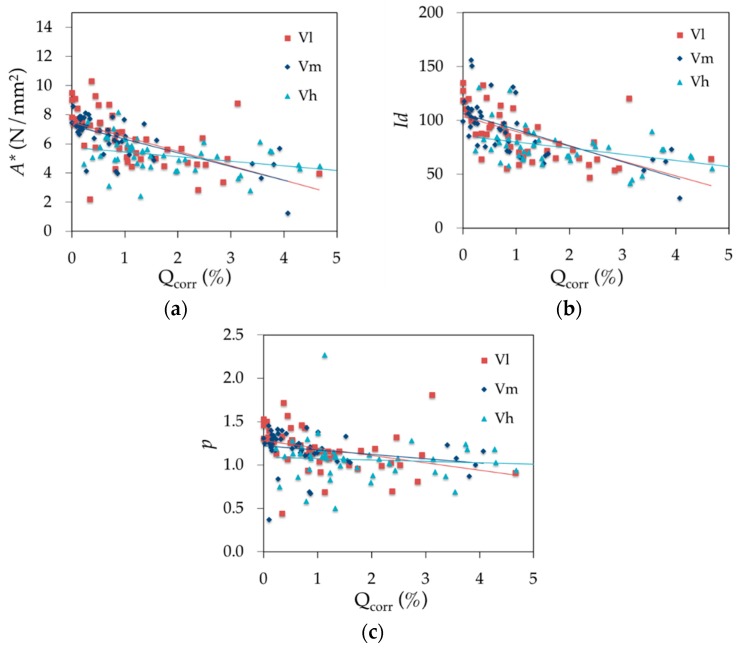
Effect of corrosion rate and loading speed on the equivalent steel concept parameters (**a**) *A**, (**b**) *Id* and (**c**) *p*.

**Table 1 materials-12-00965-t001:** Requirements of the EHE-08 code for B500SD steel.

*R*_e_ (MPa)	*R*_m_ (MPa)	*R* _m_ */R* _e_	*A*_gt_ (%)	*A*_u,5_ (%)
500	575	1.15 ≤ *R*_m_/*R*_e_ ≤ 1.35	≥7.5	≥16

**Table 2 materials-12-00965-t002:** Values of the equivalent steel parameters obtained with EHE-08 high ductility steel requirements.

Equivalent Steel Concept	Cosenza (*p*)	Creazza (*A**) (MPa)	Ortega (*Id*)
Normative EHE-08	0.82	3.87	63.65

**Table 3 materials-12-00965-t003:** Composition of concrete.

Material	Value
Cement (kg/m^3^)	290
Water (kg/m^3^)	174
Cement/water	0.6
Gravel (kg/m^3^)	1110
Sand (kg/m^3^)	760
CaCl_2_ (% cement weight)	3

**Table 4 materials-12-00965-t004:** Range of the mechanical and ductility parameters of tensile tests run at three loading speeds *V*_l_ (low loading speed), *V*_m_ (medium loading speed) and *V*_h_ (high loading speed).

Loading Speed	Values	*R*_e_ (*R*_m_) MPa	*R* _m_ */R* _e_	*A*_gt_ (%)	*A*_u,5_ (%)	*Id*	*P*	*A** (N/mm^2^)
*V* _l_	Min.	429.6(518.8)	1.07	6.05	23	47.0	0.4	2.2
	Max.	651.8(726.4)	1.21	17.29	34	135.1	1.8	10.3
*V* _m_	Min.	449.0(498.4)	1.10	3.7	7	27.8	0.4	1.3
	Max.	626.5(620.40)	1.21	14.8	35	369.0	1.5	8.6
*V* _h_	Min.	206.0(283.6)	1.03	5.2	19	41.6	0.5	2.4
	Max.	618.6(688.9)	1.4	26.4	34	205.2	2.3	8.2

**Table 5 materials-12-00965-t005:** Range of yield strength and tensile strength of tensile tests run at three loading speeds and the percentage of strength reduction.

Loading Speed	*Q*_corr_ (%)	*R*_e_ (*R*_m_) MPa	Strength Reduction Rates *(%)*
*V* _l_	0	531.7 (612.6)	–
	1	505.9 (587.3)	5%
	2	483.0 (570.3)	8%
	3	449.4 (534.9)	14%
	4	458.0 (541.9)	13%
*V* _m_	0	544.4 (626.0)	–
	1	511.5 (594.7)	6%
	2	489.5 (577.1)	9%
	3	449.0 (542.7)	16%
	4	449.4 (498.4)	19%
*V* _h_	0	547.3 (619.8)	–
	1	509.1 (591.0)	6%
	2	497.1 (574.6)	8%
	3	453.0 (544.8)	15%
	4	454.4 (533.3)	16%

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
