# Peer review of "Influence of the Loading Speed on the Ductility Properties of Corroded Reinforcing Bars in Concrete"

_materials, 2019, doi:10.3390/ma12060965_

Round 1
Reviewer 1 Report
Rebars corrosion science is a relatively active field in civil engineering as rebars bearing capacity and plasticity are of critical importance regarding the maximum solicitations a structure can withstand. Many studies have shown that the rebars mechanical properties are greatly influenced by their degradation by chlorides in marine environment because their properties significantly due to their corrosion. Laboratory test can be performed to estimate the effect of corrosion on rebars and generally consist in accelerated degradation and subsequent tensile loading.
This manuscript presents the influence of loading speed on the ductility properties of corroded reinforcing bars in concrete. This subject may be of interest to the readers as many mechanical properties depend on the loading speed. Moreover this manuscript presents a very extensive on 144 bars which adds to its interest. Globally the introduction provides a good background on the subject though some information should be added on the dependence of mechanical properties of rebars or other civil engineering materials regarding loading speed. Materials and methods are correctly presented. However the results and the associated discussion must be analyzed more carefully using statistical indicators (and that way the authors could fully get the advantage of their large dataset). Without any statistical analysis as for now, the interpretation of the results may be very questionable. The authors should use statistical analysis such as ANOVA test or Welsh t-test before exposing conclusions which are not obvious. For example, about figure 7, the authors explain l 238-239: ‘It can be observed that the deformation increases when the loading speed increases’. Due to the data points dispersion (see fig 7) for all corrosion states, it may be a misleading conclusion. And the box plots (fig 8) are very close one to each other. Without statistical analysis some of the conclusions are questionable, especially conclusion 6 which may be one of the most important as it is reported in the abstract: ‘with corrosion rates as low as 1% there is a change in the stress-strain curve so that in some cases the yield plateau disappears and the steel behaves as a cold-formed steel’
Other minor remarks should be addressed lately:
- Please check for typo errors. For example l86 ‘silica sand’, l108 a dot in the middle of a sentence, l 145 kN/s, l 186 a dot again, l 298 a dot, l 302 remove the patent section
- 53-54: the sentence ‘If this occurs in a constant homogeneous trend’ may be reformulated
- L 130: ref 40 is missing
- L 139: explain d, d0 and d1
- L 144: maybe it should be clearer to talk about ‘speed scanario’ as each of them are composed of two speeds
- L 153: please refer to eq 5 to 7
- Table 3: please stay consistant using P, p or rho
- L 180: you should maybe place this sentence before the tables
- L 217 to 224: please give examples supporting this discussion using values
- Figure 4 to 6: how do you know this curves are representatives?
- L 246-247: is this result significant again?
- Figure 8: please explain which values are represented on your box plots (median or mean value?...)
- Figure 11 is not analyzed in the text
- Figure 12: please indicate the core and corrosion products
- L 278: do you mean a slight increment of fs/fy is observed with corrosion?
- Conclusion section: please add a sentence to introduce your findings
Author Response
Rebars corrosion science is a relatively active field in civil engineering as rebars bearing capacity and plasticity are of critical importance regarding the maximum solicitations a structure can withstand. Many studies have shown that the rebars mechanical properties are greatly influenced by their degradation by chlorides in marine environment because their properties significantly due to their corrosion. Laboratory test can be performed to estimate the effect of corrosion on rebars and generally consist in accelerated degradation and subsequent tensile loading.
This manuscript presents the influence of loading speed on the ductility properties of corroded reinforcing bars in concrete. This subject may be of interest to the readers as many mechanical properties depend on the loading speed. Moreover this manuscript presents a very extensive on 144 bars which adds to its interest. Globally the introduction provides a good background on the subject though some information should be added on the dependence of mechanical properties of rebars or other civil engineering materials regarding loading speed. Materials and methods are correctly presented. However the results and the associated discussion must be analyzed more carefully using statistical indicators (and that way the authors could fully get the advantage of their large dataset). Without any statistical analysis as for now, the interpretation of the results may be very questionable. The authors should use statistical analysis such as ANOVA test or Welsh t-test before exposing conclusions which are not obvious. For example, about figure 7, the authors explain l 238-239: ‘It can be observed that the deformation increases when the loading speed increases’. Due to the data points dispersion (see fig 7) for all corrosion states, it may be a misleading conclusion. And the box plots (fig 8) are very close one to each other. Without statistical analysis some of the conclusions are questionable, especially conclusion 6 which may be one of the most important as it is reported in the abstract: ‘with corrosion rates as low as 1% there is a change in the stress-strain curve so that in some cases the yield plateau disappears and the steel behaves as a cold-formed steel’
The authors acknowledge the kind comments of the reviewer and agree with him. Nevertheless, being quite interesting the ANOVA test, it is out of the scope of this paper. For the sake of simplify the reading of the text, new wording dealing with the results of the tables three to six has been added, and these tables (three to five) moved to an Annex. Figure seven has been removed and associated comments. The new wording deals with the Figures 8 and 9 (new numbering). The conclusion 6 is based on the observation of the Figures 8 and 9, and new wording has been included.
Other minor remarks should be addressed lately:
- Please check for typo errors. For example l86 ‘silica sand’, l108 a dot in the middle of a sentence, l 145 kN/s, l 186 a dot again, l 298 a dot, l 302 remove the patent section
All typos have been corrected.
- 53-54: the sentence ‘If this occurs in a constant homogeneous trend’ may be reformulated
This sentence has been removed
- L 130: ref 40 is missing
It has been included (ref [45])
- L 139: explain d, d0 and d1
The equation has been removed and included the following equation:
Au,5 = | (3) |
- L 144: maybe it should be clearer to talk about ‘speed scanario’ as each of them are composed of two speeds
The authors think that the following wording clears the “speed scenario”:
The standard speed Vm was that recommended by standard UNE-EN ISO 15630-1: 2011 [29]. A speed three times faster Vh with 11.1 kN/s and 60.3 mm/m, and a speed Vl three times slower with 1.23 kN/s and 6.7 mm/min were also used.
- L 153: please refer to eq 5 to 7
Done (new equations four to six).
- Table 3: please stay consistant using P, p or rho
Done
- L 180: you should maybe place this sentence before the tables
Done
- L 217 to 224: please give examples supporting this discussion using values
The comments to these Figures (4 to 6) have been changed
- Figure 4 to 6: how do you know this curves are representatives?
The following wording has been included:
It can be seen that with the development of corrosion, the yield strength, tensile strength and total elongation at maximum load decreased under different strain rates. In addition, the yield plateau shortened or even disappeared. In contrast, given that with high-corrosion levels an anomalous elongation of the yield plateau is produced, the loading speed has a significant influence on this area of the curve. Compared with the uncorroded rebars, the decreased yield and tensile strength of the corroded rebars were mainly caused by the reduction of fracture cross-sectional areas. The decreased total elongation and the shortened yield plateau were due to intensified stress concentrations at the corrosion pits [47].
- L 246-247: is this result significant again?
An additional wording has been included:
Figure 8shows the range of deformations (mean value) in the yield zone for corrosion rates of lower than 1% (left) and higher than 1% (right), for the three loading speeds used. It can be seen that deformation is similar for low and medium speeds regardless of the corrosion rate. At high corrosion rates, deformation is much larger for the high loading speed.is much greater for the high loading speed (as discussed above).
- Figure 8: please explain which values are represented on your box plots (median or mean value?...)
Mean value
- Figure 11 is not analyzed in the text
This is the new Figure 13 and the comments have been included
- Figure 12: please indicate the core and corrosion products
Done
- L 278: do you mean a slight increment of fs/fy is observed with corrosion?
New wording has been included:
The evolution of ratio fs/fyRm/Refor the three loading speeds is shown in Figure 12 10. A slight increment of this ratio is observed, regardless of the loading speed. Again, this can be explained by the fact that the outer martensite presents higher values of fs/fythan the one of the ferrite in the bar core. When part of the martensite disappears the proportion of ferrite in the bar cross-section increases and the ratio fs/fybecomes higher. Other reports with corrosion rates higher than the ones of this work show that the increments of ratio fs/fyare higher than the ones here reported [28].Regardless of the loading speed, the ratio has a small fluctuation near a fixed value while the corrosion ratio changed from zero to more than 4%. It is indicated that the ratio of Rm/Reis irrelevant to the average corrosion ratio which shows that the decrease of yield strength is induced by the decrease of effective cross-section area of the steel bars [9,21].
- Conclusion section: please add a sentence to introduce your findings
A new item has been included:
7) Finally, the research significance lies in the assessment of the influence of the loading speed at which the tensile test is performed for the reinforcement bars that largely depends of the ductility criteria used.
The authors really appreciate the valuable comments of the reviewer, that help to improving the quality of the paper
Reviewer 2 Report
The present manuscript is under the scope area of Materials journal. However, there are some major issues which need to be rectified by the authors.
1. Manuscript is not properly written and it shows confusion for the readers.
2. Introduction is very limited and objective of manuscript is not clear.
3. Chemistry of studied steel rebars must be provided by the authors. Role of chemistry is very important in this study.
4. Why authors have separated the title of result and discussion? In result section only table is provided. It is better to write title as “Results and Discussion”.
5. How did authors consider whole rebar to calculate percentage of weight loss (equation 1) while some portion of it not embedded in concrete? It means taped portion of rebar is not participated in corrosion while in Pi and Pc it plays role. Thus, I think this portion is totally wrong. If correct then authors need to explain with proper reference.
6. Manuscript is scientific not sound and very weak thus it needs to be thoroughly revised.
7. The discussion of obtained results are superficial.
8. I suggest to authors to take help of English editing service to improve the language of manuscript.
Author Response
1. Manuscript is not properly written and it shows confusion for the readers.
The authors understand the comment and believe that the wording has been significantly improved with the modifications made. In addition, they have modified the symbols of the parameters in order to ease the comprehension of the paper.
2. Introduction is very limited and objective of manuscript is not clear.
The authors believe that improving the introduction enhances the paper. Therefore, they have widened it with additional paragraphs and modified some wording in order to clarify the main objective of the paper. The final wording is the following:
Corrosion of steel rebars is one of the most common deterioration mechanisms identified in reinforced concrete structures. Such corrosion, whether induced by carbonation, chlorides or another attack, affects the overall serviceability and durability of the structure with consequences such as a reduction of the effective cross-section of the steel rebars, cracking and spalling of the concrete cover and the degradation of bond strength [1-3].
In the case of chloride corrosion, the attack is mostly local and commonly known as pit corrosion. These chlorides can be found in sea water, industrial wastewater, and, among others, deicing salts [4]. Corrosion occurs after the depassivation of the alkaline barrier when sufficient oxygen and moisture are available. Then the passive film is locally destroyed and a process of local corrosion is initiated. The crucial amount of chlorides needed to interrupt the passive cover is 0.4-1% by mass of cement as an appropriate chloride threshold [5].
Over the last years more and more qualitative and quantitative research has been carried out about the reduction of the steel bars effective cross-section areas through tensile tests run by chloride corrosion.The Spanish Structural Concrete Code EHE-08 and the Eurocode EC-2 [6-7] require limited values in the mechanical properties of high-ductility steel in terms of both strength and strain. The rebar strength has a significant influence on the structural strength of concrete reinforced members and the codes require minimum values for the steel yield strength and maximum tensile strength. Additionally, due to the consideration of dynamic and seismic actions, is also required toconsideration ofconsiderproperties in relation to the steel ductility is also required. It is essential that an effective method to reflect the relationship between the mechanical and ductility property of steel bars and the corrosion be found [8-11].
One way in which ductility can be considered is in relationship with the fracture energy that is the area covered by the strain-stress curve. This energy depends on the plastic deformation capacity of steel up to breaking point. The higher the area, the higher is the capacity of steel to dissipate energy under dynamic loads. Also In addition, for dynamic and impact loads it is important the speed at which the load is applied is important.
In reinforced concrete members, the concrete cover provides protection to the rebars both physical and chemical. The alkaline environment of concrete [3-4] protects the rebars against the corrosion. In these conditions, the presence of enough humidity triggers the steel corrosion process. Three consequences take place when the steel bars start corroding: i) Loss of steel material converted in rust reducing the bar section area and the ductility [5-8] ii) cracking and spalling of the concrete cover and iii) loss of bonding between concrete and steel bar that reduces the efficiency of the reinforced concrete system [9-10].
The Codes EHE-08 and EC-2 require that the steel bars meet ductility properties based ontotal elongation at maximum force (Agt) and the ratio between tensile strength and yield strength (Rm/Re)maximum strain (εmax) and the ratio between maximum strength and the elastic limit (fs/fy). Table 1 shows the limits of those parameters as required by the EHE-08 code in order to qualify the steel as of high ductility according to standard UNE 36065:2011 [11]. In addition, although the percentage elongation after fracture (Au,5) is included in this code, it is not considered in other codes. Also, the average strain value assessed over a length of 5 bar diameters (ε5) is included in this code although is not considered in other codes.
Table 1. Requirements of the EHE-08 code for B500SD steel.
Re(MPa) | Rm(MPa) | Rm/Re | Agt(%) | Au,5(%) |
500 | 575 | 1.15 ≤ Rm/Re≤1.35 | ≥7.5 | ≥16 |
The conventionalapproach to steel rebar corrosion considers a reduction in the area of the bar section proportional to the degree of corrosion. Most of the publishedworks [13-17] report systematic reduction of strength and strain at maximum load when the degree of corrosion increases. If this occurs in a constant homogeneous trend, theloss of strength becomes proportional to the loss of bar section area. Recent studies [18] have shown that corrosion takes place in local spots of the bar surface (pitting), a weakening of strength occurs at these spots (a notch effect), and the bar strength falls under the minimum values required by the codes, even with very small degrees of corrosion. Nevertheless, the reduction of strain is greater than the loss of strength in the bar [19].
With low levels of corrosion the loss of strength is also lowand: this meansthat the structural elements can still meet their resistance function, but thoughthe reduction of strain may not meet the minimum values required in Table 1 to ensure enough ductility. Previous studies [20-21] have shown that the ratio Rm/Reremains constant with the increase of the corrosion level. This means that the steel may amply meet the Rm/Rerequirement but not the requirement ofAgt.Previous studies [17] show that the ratio fs/fyremains constant with the increase of the corrosion level. That means that the steel may amply meet the fs/fy requirement but not the requirement ofεmax.
In these cases, the use of theequivalent steel concept as a ductility criterion based on both Rm/Reand Agtmay be highly useful [22-23] fs/fyand εmaxcan be very useful. Table 2 shows the minimum values obtained with the EHE-08 requirements in application of the equivalent steel formulas [24-28] as proposed by Cosenza (p), Creazza (A*) and Ortega (Id).
Table 2. Values of the equivalent steel parameters obtained with EHE-08 high ductility steel requirements.
Cosenza (ρ) | Creazza (A*) | Ortega (Id) | |
Normative EHE-08 | 0.82 | 3.87 | 63.65 |
Moreover, although the standard test procedure of tensile tests of reinforcing bars ISO 15630-1:2010 [29] is remarkably complete, no recommendation regarding the loading speed for the tensile test is set. This might have a significant impact on the test results. For this reason, in this study the variations of the mechanical properties of steel rebars as a function of the degree of corrosion and the loading speed applied in the tensile tests are reported. For such a purpose, 144 12-mm diameter high-ductility steel rebars, named B500SD, this work the variations of the mechanical steel properties as a function of the degree of corrosion and of the increasing speed of the applied load are reported. For that purpose, 144 bars of 12 mm diameter of reinforcing high ductility steel, named B500SDwere tested in tension after an accelerated corrosion treatment, whenembedded in NaCl contaminated concrete. The results showed that the influence of high-speed loading is significant if only the Codes EHE-08 and EC-2 are used. In order to establish a better relationship between the tensile test and the minimum effective cross-sectional area, the use of the equivalent concept is required.
3. Chemistry of studied steel rebars must be provided by the authors. Role of chemistry is very important in this study.
The authors agree in the key role of chemistry in this study. Thus, they have incorporated the following paragraphs in the introduction:
Corrosion of steel rebars is one of the most common deterioration mechanisms identified in reinforced concrete structures. Such corrosion, whether induced by carbonation, chlorides or another attack, affects the overall serviceability and durability of the structure with consequences such as a reduction of the effective cross-section of the steel rebars, cracking and spalling of the concrete cover and the degradation of bond strength [1-3].
In the case of chloride corrosion, the attack is mostly local and commonly known as pit corrosion. These chlorides can be found in sea water, industrial wastewater, and, among others, deicing salts [4]. Corrosion occurs after the depassivation of the alkaline barrier when sufficient oxygen and moisture are available. Then the passive film is locally destroyed and a process of local corrosion is initiated. The crucial amount of chlorides needed to interrupt the passive cover is 0.4-1% by mass of cement as an appropriate chloride threshold [5].
4. Why authors have separated the title of result and discussion? In result section only table is provided. It is better to write title as “Results and Discussion”.
The authors appreciate the comment and have grouped the results and discussion sections into one. In addition, the tables have been placed in Annex I in order to ease the reading of the paper.
5. How did authors consider whole rebar to calculate percentage of weight loss (equation 1) while some portion of it not embedded in concrete? It means taped portion of rebar is not participated in corrosion while in Pi and Pc it plays role. Thus, I think this portion is totally wrong. If correct then authors need to explain with proper reference.
The authors only considered for the calculations the portion of rebar that participates in corrosion, deducting the percentage of rebar that is not embedded. This gravimetric procedure has been widely accepted and is described in detail in reference [41]. In any event, the authors believe that it was worth clarifying the comment and have included new wording regarding this procedure:
The degree of corrosion or corrosion level (Qcorr) was measured by the gravimetric procedure of weighing the bars after the full cleaning of all corrosion products. The gravimetric cross-sectional loss of the corroded steel reinforcement was deduced by Equation (1) [41].
Qcorr= | (1) |
where Qcorris the corrosion degree of the steel reinforcement in percentage and and are the mass of initial reinforcement corresponding to the portion of rebar that participates, respectively, in corrosion and the residual mass of the corroded steel reinforcement portion.
The reference 41 has also been completed as follows:
41. Moreno Fernández, Esther. "Corrosión de armaduras en estructuras de hormigón: Estudio experimental de la variación de la ductilidad en armaduras corroídas aplicando el criterio de acero equivalente."Doctoral Thesis, Universidad Carlos III de Madrid (2008).
6. Manuscript is scientific not sound and very weak thus it needs to be thoroughly revised.
The authors believe that with the improvements performed in the introduction, the modifications of the structure, the new wording, figures, tables, as well as the linguistic revision the paper have significantly improved it not only in scientific terms but also in clarifying the research significance.
7. The discussion of obtained results are superficial.
The authors understand the comment and have widened the discussion in order to supply deeper analysis of the results. The new wording is the following:
rcentage of bar specimens that meetdifferent ductility criteria in tensile strength tests run at different loading speeds: low Vl, medium Vm and high Vh.
As expected, when using EHE-08 criteria, the loading speed is important with high levels of corrosion. Even with corrosion rates of up to 1% there is a significant difference for high loading speed (Vh) as compared with low and medium speeds (Vl, Vm). In addition, for corrosion rates of up to 1% all equivalent steel criteria are met for low and medium loading speeds. For high-speed loading the criteria offered by Creazza are scarcely met, although fulfilment is frequent for Cosenza and Ortega criterion. With corrosion rates of higher than 1%, fulfilment of EHE-08 ductility criteria was low, less so for Cosenza criterion and high for the Creazza and Ortega criterion. Thus, in general, the three equivalent steel concepts offered by Cosenza, Creazza and Ortega serve as useful criteria for high loading speeds and corrosion rates under 1%. Given that more that 90% of the bar specimens meet the ductility criteria, the concept is quite advantageous in assessing structural ductility with corroded reinforcement. Regardless of the loading speed considered, the EHE-08 ductility requirements are met by more than 90% of the bar specimens for corrosion rates of up to 1%, though only 20% of the specimens meet such requirements when the corrosion rate is higher than 1%. The main reason is the systematic reduction of the total elongation at maximum load (Agt) when it increases the corrosion rates [30] up to values that fail the minimum ones required by EHE-08.
Summaries of representative strain-stress curves are plotted in Figure 5, Figure 6and Figure7 for, respectively, each low, medium and high loading speed, respectively. It can be seen in all of them that strains at the elastic limit and at the maximum load (εmax) decrease when the corrosion rate increases. This effect is more pronounced for εmax. The numbers in % indicate the corrosion level of the rebar.
If Figure 6 is compared with Figures 4 and 5 it can be observed that the strain in the yield plateau (after the elastic limit is reached) is higher in bar specimens tested at high loading speed as compared with the specimens tested at low and medium loading speeds.
It can be seen that with the development of corrosion, the yield strength, tensile strength and total elongation at maximum load decreased under different strain rates. In addition, the yield plateau shortened or even disappeared. In contrast, given that with high-corrosion levels an anomalous elongation of the yield plateau is produced, the loading speed has a significant influence on this area of the curve. Compared with the uncorroded rebars, the decreased yield and tensile strength of the corroded rebars were mainly caused by the reduction of fracture cross-sectional areas. The decreased total elongation and the shortened yield plateau were due to intensified stress concentrations at the corrosion pits [47].
Figure 5. Representative summary of the strain-stress curves of the 48 bar specimens tested at low speed Vl for increasing corrosion rates.
Figure 6.Representative summary of the strain-stress curves of the 48 bar specimens tested at medium (standard) speed Vm for increasing corrosion rates.
Figure 7. Representative summary of the strain-stress curves of the 48 bar specimens tested at high speed Vh for increasing corrosion rates.
In Figure 7 the total deformation of the bars in the yield zone is plotted for the three loading speeds as a function of the corrosion rate Qcorr. It can be observed that the deformation increases when the loading speed increases.
Figure 7. Deformation of bar specimens in the yield zone as a function of the corrosion rate for each of the three loading speeds.
Figure 8shows the range of deformations in the yield zone for corrosion rates of lower than 1% (left) and higher than 1% (right), for the three loading speeds used. It can be seen that deformation is similar for low and medium speeds regardless of the corrosion rate. At high corrosion rates, deformation is much larger for the high loading speed.is much greater for the high loading speed (as discussed above).
Figure 8. Deformation of bar specimens in the yield zone for corrosion rates under 1% (left) and over 1% (right) and for the three loading speeds Vl, Vm and Vh.
Regardless of the corrosion level, in Figure 9 it is possible to see that the deformation of bar specimens in the yield zone showed less dispersion for low and medium speeds in comparison with high speed.
Figure 9. Deformation of bar specimens in the yield zone as a function of the loading speed Vl (left), Vm (centre) and Vh (right).
The evolution of the mechanical properties obtained in the tensile tests as a function of the corrosion rate is plotted in Figures 9, 10 and 11. Colours allow distinction of the loading speed.
The evolution of the mechanical properties obtained in the tensile tests as a function of the corrosion rate can be observed in Figure 10. It shows the average cross-section diameter of the specimen after the corrosion process. Each colour denotes the loading speed. The figure also shows that the yield strength, tensile strength and total elongation at maximum force, regardless of the loading speed, decrease with the increase in corrosion level. The dispersion of the results occurs only with high corrosion levels. Therefore, the loading speed in this test program seemed to have no significant effect on either strength at a low level of corrosion.
Figure 9.Effects of the corrosion rate and the loading speed on the tensile strengthfs(left) and the elastic limit fy(right).
Figure 10. Effects of the corrosion rate and the loading speed on the ratio fs/fy.
Figure 11. Effects of the corrosion rate and the loading speed on the strain at maximum load εmax.
Figure 10.Effects of the corrosion rate and loading speed on the tensile strengthRm, yield strength Reand total elongation at maximum force Agt.
The values of the tensile strength fsand the elastic limit fyof Figure 9 have been obtained by dividing the acting load by the average cross-section area of the specimen σresafter the corrosion process. If the corrosion would havehadbeen uniform along the bar, the adjusting trend lines wouldhave been horizontal. However, the lines aredecreasefor all loading speeds [48]. This isdue to the factbecausethat thecorrosion is not homogeneous in the bar surface, but occurs in a series of pitting spots typical for chloride corrosion of steel [49-50]. In these spots, the cross-section area of the bar is smaller than the average in the test results (see Figure 11). Additionally, thecorrosion takes place in the outer thickness of the bar surface composed by martensite, a metallographic material produced by the rolling mill when the bar was fabricated. Martensite has higher strength properties (fs, fyRm, Re) than the ferrite composing the internal core of the bar. The destruction of part of this stronger outer layer explains the reduction of the average strength values in the bar cross-section.
Figure 11. Microscopy image of the corroded surface (left) of bar specimen B087M with Qcorr= 1.07% and the cross-section (right).
The evolution of ratio fs/fyRm/Refor the three loading speeds is shown in Figure 12 10. A slight increment of this ratio is observed, regardless of the loading speed. Again, this can be explained by the fact that the outer martensite presents higher values of fs/fythan the one of the ferrite in the bar core. When part of the martensite disappears the proportion of ferrite in the bar cross-section increases and the ratio fs/fybecomes higher. Other reports with corrosion rates higher than the ones of this work show that the increments of ratio fs/fyare higher than the ones here reported [28].Regardless of the loading speed, the ratio has a small fluctuation near a fixed value while the corrosion ratio changed from zero to more than 4%. It is indicated that the ratio of Rm/Reis irrelevant to the average corrosion ratio which shows that the decrease of yield strength is induced by the decrease of effective cross-section area of the steel bars [9,21].
Figure 12. Effects of the corrosion rate and the loading speed on the ratio Rm/Re.
Figure 13 11shows the evolution of the three ductility parameters based on the steel equivalent concept as a function of the corrosion rate for the three loading speeds. All parameter values decrease when the corrosion rate increases regardless of the loading speed. Parameters pand A*evolve similarly for low and medium loading speeds (parallel lines).Parameters evolve similarly for low and medium loading speeds (parallel lines). However, the degree of scatter with high-speed loading would not provide accurate conclusions.
Figure 13.Effect of corrosion rate and loading speed on the equivalent steel concept parameters A*, Idand p.
8. I suggest to authors to take help of English editing service to improve the language of manuscript.
The wording of the paper has been thoroughly revised by a native speaker with expertise in the field.

Reviewer 3 Report
There are already so many existing literature deals with the strength of corroded steel bar. So, authors need to highlight what is the new information in this paper. Therefore, this paper needs major revision as there are so many things which authors didn’t consider in the current version of the paper. Please revise the manuscript as per the suggestion below:
1. Accelerated test doesn’t replicate the real life scenario as different researchers use different acceleration method. Can authors link the accelerated corrosion method adopted here with the real time corrosion? You may refer to the following paper: A novel link of the time scale in accelerated chloride-induced corrosion test in reinforced SHCC. Construction and Building Materials 2018, 167, 15-19.
2. After corrosion testing were the bars clean before they used in tensile test again? How the Pc values were determined in Eq 1?
3. In Eq 2, is the corroded length Lc is same as the embedded length of steel in concrete slab? Was the corrosion distributed to the whole embedded length of bar? Please discuss it as this information is very important. Also, add some photos of corroded bars in section 2.2. Figure 12 is not sufficient enough to understand the corrosion scenario in the bars. The following paper will be helpful to read and add as it discussed corrosion, cleaning of steel bar and testing them in tension. Crack formation and chloride induced corrosion in reinforced strain hardening cement-based composite (R/SHCC). Journal of Advanced Concrete Technology 2014, 12, 340-351.
4. The visibility is not good enough for most of the figures 4-13. Please improve it.
5. Please define, what is meant by d, do, d1 in Eqs 3&4. How did you measure the cross-sectional area of corroded bar? Was the corrosion uniform in the bar? If not how you did you calculate corroded cross-sectional area of the bar and also tensile strength loss, please explain it.
6. You could also report mass loss, pitting depths, etc. which are considered to be the important for corrosion of steel.
7. The literature review part also needs to improve. There are so many very old references are added which may need to revise with new references as suggested below.
Broomfield, J. P. (2007). Corrosion of steel in concrete understanding, investigating and repair. Book 2nd edition, Taylor & Francis, USA & Canada.
https://doi.org/10.3390/s16010015
https://doi.org/10.1007/s40069-017-0205-8
https://doi.org/10.1007/s40069-013-0054-z
Author Response
There are already so many existing literature deals with the strength of corroded steel bar. So, authors need to highlight what is the new information in this paper. Therefore, this paper needs major revision as there are so many things which authors didn’t consider in the current version of the paper. Please revise the manuscript as per the suggestion below:
1. Accelerated test doesn’t replicate the real life scenario as different researchers use different acceleration method. Can authors link the accelerated corrosion method adopted here with the real time corrosion? You may refer to the following paper: A novel link of the time scale in accelerated chloride-induced corrosion test in reinforced SHCC. Construction and Building Materials 2018, 167, 15-19.
The authors appreciate the comment and have included new wording and the reference suggested (34) in order to improve the paper. Various research studies can be found in the literature which use this acceleration method (most use current densities ranging up up 200mA / cm2[37-38]). In the case of this study, the current density was 10 mA / cm2, which is much lower than indicated. In addition, Andrade [36] indicated that values above 10mA / cm2have almost never been recorded in real size structures. In addition, Maaddaway et al. concluded that more research work is required to investigate the ability of Faraday's law to predict the percentages of the mass at the corrosion level than 7.27% mass loss. However, in this research as that degree of corrosion was not reached the error in that area was limited. In such a sense, the authors have included the following new wording as clarification:
This accelerated corrosion test is only valid if the intensity remains rather low (< 200 μA/cm2) with respect to Faraday’s law. Otherwise a significant increase of strain response, and consequently the crack width, will occur. Densities can reach 200μA/cm2, with increasingly more appearing that will undermine accuracy of the experiment. These current values are broadly explained by Andrade [36], Maaddawy [37] or Suvash [38]. In order to obtain distinct corrosion levels, the current was disconnected at different ages after cracks appeared in the concrete slabs (Figure 2).
2. After corrosion testing were the bars clean before they used in tensile test again? How the Pc values were determined in Eq 1?
The bars were cleaned following the standard ASTM G1-03 (2017)e1 [39]. Regarding the corrosion level, the gravimetric procedure describe in reference 41 was followed. As the authors understand that clarifying this point is of high interest, they have improved the text as follows:
“Once the corrosion process was over, the slabs were demolished, the oxide and cement of each bar surface were mechanically cleaned by a brush according to standard ASTM G1-03 (2017)e1 [39]. Then, if such a treatment could not eliminate all the corrosion products, the oxide was cleaned by a chemical process by immersing the bars in a bath for 10 minutes. It was then rinsed with ethanol and water and dried according to standard ISO 8407 [40]. As it can be observed in Figure 3, the corrosion was extended throughout the whole surface of embedded bars.”
3. In Eq 2, is the corroded length Lc is same as the embedded length of steel in concrete slab? Was the corrosion distributed to the whole embedded length of bar? Please discuss it as this information is very important. Also, add some photos of corroded bars in section 2.2. Figure 12 is not sufficient enough to understand the corrosion scenario in the bars. The following paper will be helpful to read and add as it discussed corrosion, cleaning of steel bar and testing them in tension. Crack formation and chloride induced corrosion in reinforced strain hardening cement-based composite (R/SHCC). Journal of Advanced Concrete Technology 2014, 12, 340-351.
The authors appreciate the valuable comment and have included the reference [38], new wording and a picture, as suggested by the reviewer. The use of Eq. 2 has been adapted from reference [42-43] by using the residual diameter value. Lc was the corroded length of the bar, with it being the same as the embedded length. As can be observed in the new Figure 3, the corrosion was distributed throughout the embedded bar. Moreover, Figure 2 has been improved to help readers to understand the corrosion scenario in the bars. The new wording and figure can be read below:
Figure 3. Rebars after the first treatment according to standard ASTM G 190-06.
The corrosion level of each bar was quantified by a variableQmaxassessed by gravimetric procedures. weighing the bars after the full cleaning of all corrosion products and assuming that the loss of steel has been uniform in the corroded bar length Lc that is the equation:
Qcorr = , | (1) |
Where Qcorris the percentage of weight loss and Piand Pcare the initial (previous to the slab fabrication) and final weight of the bar, respectively.
The degree of corrosion or corrosion level (Qcorr) was measured by the gravimetric procedure of weighing the bars after the full cleaning of all corrosion products. The gravimetric cross-sectional loss of the corroded steel reinforcement was deduced by Equation (1) [41].
Qcorr= | (1) |
where Qcorris the corrosion degree of the steel reinforcement in percentage and and are the mass of initial reinforcement corresponding to the portion of rebar that participates, respectively, in corrosion and the residual mass of the corroded steel reinforcement portion.
The residual value of the bar cross-section area as average in the length Lccan be determined by the equation:
Sres = | (2) |
Where Sresis the equivalent residual section (cm2), Lcis the corroded length of the bar (cm) and 7.85 is the specific weight of steel (g/cm3).
In order to calculate the residual diameter value of the corroded bar cross-section, the residual mass of the corroded steel reinforcement is used and determined following the equivalent section definition [42-43].
Thus, the residual diameter of the corroded bars was computed by Equation (2) as follows:
= | (2) |
where Lcis the corroded length of the bar (cm) and 7.85 is the specific weight of steel (g/cm3).
4. The visibility is not good enough for most of the figures 4-13. Please improve it.
The quality of all the figures have been improved.
5. Please define, what is meant by d, do, d1 in Eqs 3&4. How did you measure the cross-sectional area of corroded bar? Was the corrosion uniform in the bar? If not how you did you calculate corroded cross-sectional area of the bar and also tensile strength loss, please explain it.
The authors understand that some of the nomenclature could be confusing and have modified all the parameters in Eq. 3 in order to ease comprehension. The measurement was performed following standard UNE EN ISO 6892-1: 2017. It can be seen in Figure 3 that corrosion existed in the whole bar. Concerning Eq.4, as the authors believe that it could create confusion it has been removed from the text given that it was directly measured by means a gauge of class 2 following standard ISO 9513:2012. The obtaining of the corroded cross-sectional area has been explained in detail as can be read in the final wording:
Before the test, for the manual determination of the elongation after fracture (Equation (3), the entire bar received fine marks with multiples of 5mm following the UNE EN ISO 6892-1: 2017 standard [45]. In addition, for the total elongation at maximum force a class two strain gauge, according to the ISO 9513:2012 standard, was used [46].
Au,5 = | (3) |
where Au,5is the permanent elongation of the gauge length expressed as a percentage of the original gauge length,Luis the final gauge length after fracture and L0is the original gauge length.
6. You could also report mass loss, pitting depths, etc. which are considered to be the important for corrosion of steel.
The authors understand the comment. As regards the mass loss, they believe that this value is implicit in the degree of corrosion or corrosion level and Table 3, Table 4 and Table 5 were already extensive in terms of data. In the case of pitting depths, it was out of the scope of this study although it would have been of great interest.
7. The literature review part also needs to improve. There are so many very old references are added which may need to revise with new references as suggested below.
The authors have improved the literature review and have included several new references in addition to those supplied by the reviewer [1, 2, 3, and 35]. They also thank the reviewer for improving the paper.

Reviewer 4 Report
Major comments:
What is the novelty of this study? What does it add to the current literature? The findings in this study are not new.
Several grammatical and English errors can be seen throughout the manuscript.
Please add the main findings of the research to the Abstract.
Lines 35-36: Actually, concrete only provided physical protection, but this physical protection can also prevent chemical damage. This sentence should be revised.
Line 36-37: it is not only the alkaline environment feature of the concrete that protects the rebars. If it was only the alkaline environment, there would never be a corrosion in the concrete. Several other features of the concrete also play the role.
In the introduction section, it is required to add some information regarding what is done in the paper. Then, the main findings can be stated.
Minor comments:
Line 86: “san”?
Line 112: Please change “Where” to “where” and remove the indentation. Please do the same where it applies.
Line 186: Change “.” To “,”
Line 295: “loadiand”? “g”?
Author Response
What is the novelty of this study? What does it add to the current literature? The findings in this study are not new
The authors believe that with the modifications made to the wording and structure of the paper, the significance of the research has been clarified. With the huge amount of rebars used it was possible to verify that a significant number of rebars, even with low levels of corrosion, are rejected with the requirements of the codes EHE-08 and EC-2. Moreover, the value of loading speed also affected the results, which is neglected in the majority of codes. In those cases, with high loading speeds, the use of the equivalent concept is imperative to establish a better relationship between the tensile test and the minimum effective cross-section area.
Several grammatical and English errors can be seen throughout the manuscript.
The wording of the paper has been thoroughly revised by a native speaker with expertise in the field.
Please add the main findings of the research to the Abstract.
The Abstract has been modified in order to highlight the findings in the study. The new wording is the following:
Abstract: In this work 144 reinforcing bars of high-ductility steel named B500SD were subjected to an accelerated corrosion treatment and then tested under tension at different loading speeds in order to assess the effect of corrosion on the ductility properties of the rebars. Results showed that the bars with a corrosion level as low as the one reducing the steel mass by 1% gave rise to a significant degradation on the ductility properties when a high loading speed was applied in tensile tests. In that case, the equivalent steel concept is useful to reduce the destabilising effect. Thus, the research significance lies in the assessment of the influence of the loading speed at which the tensile test is performed for the reinforcement bars that largely depends of the ductility criteria used.
Lines 35-36: Actually, concrete only provided physical protection, but this physical protection can also prevent chemical damage. This sentence should be revised.
Line 36-37: it is not only the alkaline environment feature of the concrete that protects the rebars. If it was only the alkaline environment, there would never be a corrosion in the concrete. Several other features of the concrete also play the role.
The authors appreciate the comment and have modified the two sentences in order to clarify them. The new wording is the following:
Corrosion of steel rebars is one of the most common deterioration mechanisms identified in reinforced concrete structures. Such corrosion, whether induced by carbonation, chlorides or another attack, affects the overall serviceability and durability of the structure with consequences such as a reduction of the effective cross-section of the steel rebars, cracking and spalling of the concrete cover and the degradation of bond strength [1-3].
In the case of chloride corrosion, the attack is mostly local and commonly known as pit corrosion. These chlorides can be found in sea water, industrial wastewater, and, among others, deicing salts [4]. Corrosion occurs after the depassivation of the alkaline barrier when sufficient oxygen and moisture are available. Then the passive film is locally destroyed and a process of local corrosion is initiated. The crucial amount of chlorides needed to interrupt the passive cover is 0.4-1% by mass of cement as an appropriate chloride threshold [5].
In the introduction section, it is required to add some information regarding what is done in the paper. Then, the main findings can be stated.
The introduction section has been modified in depth and the main findings of the paper highlighted. The modified introduction can be read below:
Moreover, although the standard test procedure of tensile tests of reinforcing bars ISO 15630-1:2010 [29] is remarkably complete, no recommendation regarding the loading speed for the tensile test is set. This might have a significant impact on the test results. For this reason, in this study the variations of the mechanical properties of steel rebars as a function of the degree of corrosion and the loading speed applied in the tensile tests are reported. For such a purpose, 144 12-mm diameter high-ductility steel rebars, named B500SD, this work the variations of the mechanical steel properties as a function of the degree of corrosion and of the increasing speed of the applied load are reported. For that purpose, 144 bars of 12 mm diameter of reinforcing high ductility steel, named B500SDwere tested in tension after an accelerated corrosion treatment, whenembedded in NaCl contaminated concrete. The results showed that the influence of high-speed loading is significant if only the Codes EHE-08 and EC-2 are used. In order to establish a better relationship between the tensile test and the minimum effective cross-sectional area, the use of the equivalent concept is required.Minor comments:
Line 86: “san”?
Line 112: Please change “Where” to “where” and remove the indentation. Please do the same where it applies.
Line 186: Change “.” To “,”
Line 295: “loadiand”? “g”?
The authors also appreciate also these minor comments. They have all been revised and modified.

Reviewer 5 Report
Work may be considered for publication after the concerns below are addressed:
"The alkaline environment of concrete [3-4] protects the rebars against the corrosion. In these conditions, the presence of enough humidity triggers the steel corrosion process." This statement makes no sense. Corrosion wont happen unless the pH drops (loss of alkalinity) or you have significant chloride ingress. Please correct statement.
L54 "If this occurs in a constant homogeneous trend." I do not follow this statement. Please proofread the paper carefully - there are a few errors with language here. Other examples are "san" instead of "sand" in L86 and "Cats" in Figure 1.
L73-76: Can you clearly state how this is different from what exists in literature?
Present concrete mixture design.
I see no point in Table 3-5. This is far too much data. Show/summarize graphically.
L185-188: Much easier to see if shown in a graph.
Please indicate what the numbers in % on the figures indicate in Figure 4-6.
Figure 7: So much scatter in this data. Is there a better way to show this?
I think you need to describe the results from the figures in a bit more detail.
Figure 12 is not mentioned in the text nor is the procedure for microscopy detailed.
Please place your results in the context of other studies from literature and compare your finding with those from other studies. This is critical.
Author Response
REVISOR 5
Yes | Can be improved | Must be improved | Not applicable | |
Does the introduction provide sufficient background and include all relevant references? | ( ) | ( ) | (x) | ( ) |
Is the research design appropriate? | ( ) | (x) | ( ) | ( ) |
Are the methods adequately described? | ( ) | (x) | ( ) | ( ) |
Are the results clearly presented? | ( ) | ( ) | (x) | ( ) |
Are the conclusions supported by the results? | ( ) | ( ) | (x) | ( ) |
Comments and Suggestions for Authors
Work may be considered for publication after the concerns below are addressed:
"The alkaline environment of concrete [3-4] protects the rebars against the corrosion. In these conditions, the presence of enough humidity triggers the steel corrosion process." This statement makes no sense. Corrosion wont happen unless the pH drops (loss of alkalinity) or you have significant chloride ingress. Please correct statement.
The authors agree with the comment of the reviewer and this paragraph has been removed.
L54 "If this occurs in a constant homogeneous trend." I do not follow this statement. Please proofread the paper carefully - there are a few errors with language here. Other examples are "san" instead of "sand" in L86 and "Cats" in Figure 1.
The authors thank the comment of the reviewer and the errors have been corrected. The phrase "If this occurs in a constant homogeneous trend" has been removed.
L73-76: Can you clearly state how this is different from what exists in literature?
The accelerated corrosion test has been detailed in lines 127-203.
Present concrete mixture design.
Unfortunately, we do not have access to this data, but an additional wording has been included in the text:
River silica, sand, round gravel and cement CEM II/A-L 32.5, according to standard RC-16 [30] were the mix basic materials. The water/binder ratio was 0.6. NaCl with a concentration of 2% relative to the weight of cement was diluted in the mixing tap water in order to destroy the passive state of the reinforcing bars.
We think that the water/binder ratio is the key point
I see no point in Table 3-5. This is far too much data. Show/summarize graphically.
These tables have been moved to an Annex.
L185-188: Much easier to see if shown in a graph.
The content of Discussion section has been written again (lines 260-443).
Please indicate what the numbers in % on the figures indicate in Figure 4-6.
This aspect has been included in the new wording of the lines 260-443
Figure 7: So much scatter in this data. Is there a better way to show this?
This Figure has been removed and a new Figure 9 has been included, that shows the scatter in a better way.
I think you need to describe the results from the figures in a bit more detail.
It has been done with new wording.
Figure 12 is not mentioned in the text nor is the procedure for microscopy detailed.
It has been done (line 414).
Please place your results in the context of other studies from literature and compare your finding with those from other studies. This is critical.
It has been done with the new wording in lines 260-443
Round 2
Reviewer 1 Report
The authors took into consideration most of the reviewers’remarks and the structure of the article and the analysis of the results are more detailed and may be of interest to the readers. The authors should correct some minor details before publication:
L 176: ‘zonesat’
l 183: chose between does or did
Fig 9: please do not plot a line between the points if they represent different specimens
L 327: add a basic sentence to introduce your different findings...
p { margin-bottom: 0.25cm; line-height: 120%; }
Author Response
The authors took into consideration most of the reviewers’remarks and the structure of the article and the analysis of the results are more detailed and may be of interest to the readers. The authors should correct some minor details before publication:
L 176: ‘zonesat’
Corrected
l 183: chose between does or did
Corrected: did
Fig 9: please do not plot a line between the points if they represent different specimens
The authors think that the line (even in the case of representing various specimens) helps to the reader to see the tendency of the deformation as a function of the Qcorr. In any case, it may be removed if the reviewer thinks that it is better.
L 327: add a basic sentence to introduce your different findings...
Done: "The main conclusions can be summarised as follows:"
We would like to thank the reviewer for his/her detailed comments, which have helped us to improve the quality of the paper and to address important details.
Reviewer 2 Report
accept in present form
Author Response
We would like to thank the reviewer for his/her detailed comments, which have helped us to improve the quality of the paper and to address important details.
Reviewer 3 Report
Corrosion is also influenced by the concrete properties itself. Therefore, in section 2.1, it will nice to add the concrete compositions and strength.
Author Response
Corrosion is also influenced by the concrete properties itself. Therefore, in section 2.1, it will nice to add the concrete compositions and strength.
It has been included a Table with the concrete composition.
The mean compressive strength of concrete was 26MPa.
The authors would like to thank the reviewer for his/her detailed comments, which have helped us to improve the quality of the paper and to address important details.
Reviewer 4 Report
Thanks for addressing the comments.
Author Response
The authors would like to thank the reviewer for his/her detailed comments, which have helped us to improve the quality of the paper and to address important details.
Reviewer 5 Report
I have no further comments.
Author Response

(The authors gave the same response as above.)
